# Consumption of Tea, Alcohol, and Fruits and Risk of Kidney Stones: A Prospective Cohort Study in 0.5 Million Chinese Adults

**DOI:** 10.3390/nu13041119

**Published:** 2021-03-29

**Authors:** Han Wang, Junning Fan, Canqing Yu, Yu Guo, Pei Pei, Ling Yang, Yiping Chen, Huaidong Du, Fanwen Meng, Junshi Chen, Zhengming Chen, Jun Lv, Liming Li

**Affiliations:** 1Department of Epidemiology and Biostatistics, School of Public Health, Peking University, Beijing 100191, China; wanghan964@163.com (H.W.); fanjunning219@163.com (J.F.); yucanqing@pku.edu.cn (C.Y.); lmlee@vip.163.com (L.L.); 2Center for Public Health and Epidemic Preparedness and Response, Peking University, Beijing 100191, China; 3Chinese Academy of Medical Sciences, Beijing 100730, China; guoyu@kscdc.net (Y.G.); peipei@kscdc.net (P.P.); 4Medical Research Council Population Health Research Unit, University of Oxford, Oxford OX3 7LF, UK; ling.yang@ndph.ox.ac.uk (L.Y.); yiping.chen@ndph.ox.ac.uk (Y.C.); huaidong.du@ndph.ox.ac.uk (H.D.); 5Clinical Trial Service Unit and Epidemiological Studies Unit (CTSU), Nuffield Department of Population Health, University of Oxford, Oxford OX3 7LF, UK; zhengming.chen@ndph.ox.ac.uk; 6Liuzhou Center for Disease Control and Prevention, NCDs Prevention and Control Department, Liuzhou, Guangxi 545007, China; meng_fan_wen@126.com; 7China National Center for Food Safety Risk Assessment, Beijing 100022, China; jshchen@ilsichina.org; 8Key Laboratory of Molecular Cardiovascular Sciences, Peking University, Ministry of Education, Beijing 100191, China

**Keywords:** kidney stones, cohort study, tea consumption, alcohol consumption, fruit consumption

## Abstract

A few prospective studies have suggested that tea, alcohol, and fruit consumption may reduce the risk of kidney stones. However, little is known whether such associations and their combined effect persist in Chinese adults, for whom the popular tea and alcohol drinks are different from those investigated in the aforementioned studies. The present study included 502,621 participants from the China Kadoorie Biobank (CKB). Information about tea, alcohol, and fruit consumption was self-reported at baseline. The first documented cases of kidney stones during follow-up were collected through linkage with the national health insurance system. Cox regression was used to calculate the hazard ratio (HR) and 95% confidence interval (CI). During a median of 11.1 years of follow-up, we collected 12,407 cases of kidney stones. After multivariable adjustment, tea, alcohol, and fruit consumption were found to be negatively associated with kidney stone risk, but the linear trend was only found in tea and fruit consumption. Compared with non-tea consumers, the HR (95% CI) for participants who drank ≥7 cups of tea per day was 0.73 (0.65–0.83). Compared with non-alcohol consumers, the HR (95% CI) was 0.79 (0.72–0.87) for participants who drank pure alcohol of 30.0–59.9 g per day but had no further decrease with a higher intake of alcohol. Compared with less-than-weekly consumers, the HR (95% CI) for daily fruit consumers was 0.81 (0.75–0.87). Even for those who did not drink alcohol excessively, increasing tea and fruit consumption could also independently reduce the stone risk. Among Chinese adults, tea, alcohol, and fruit consumption was associated with a lower risk of kidney stones.

## 1. Introduction

Kidney stones are crystals formed from the supersaturation of minerals in urine and deposited in renal calyces and pelvis [1]. Kidney stone disease, one of the most common urological disorders, is associated with increased risks of hypertension, chronic kidney disease, and end-stage renal disease [2,3]. In recent decades, the incidence and prevalence of kidney stones have continued to rise worldwide [4]. According to survey data from eight countries, the annual incidence of kidney stones was about 114 to 720 cases per 100,000 people, and the prevalence was about 1.7% to 14.8% [5]. In China, the prevalence of kidney stones was approximately 5.8% to 7.5% [6,7]. Kidney stones also have a high recurrence rate, about 30% within five years of the initial onset [8]. Given the medical costs and the social burden from this disease, kidney stones should be recognized as an important health concern [9].

Previous studies have shown that fluid and dietary intake, metabolic syndrome, and other factors might play essential roles in developing kidney stones [10,11,12,13]. Tea, alcohol, and fruits are important sources of fluid and a variety of chemical components. Only a few prospective studies have examined the associations of tea, alcohol, and fruit consumption with kidney stone risk, most of which were conducted in Western populations. Findings for fruit consumption are relatively consistent; a higher average daily consumption of fruits was associated with a lower risk of kidney stones [12,14,15]. In European and American populations, the association between tea consumption and kidney stone risk was inconsistent. Some studies found a lower stone risk for tea consumers [15,16], while the another did not [17]; none distinguished the types of tea consumed. A cohort study in Shanghai of China found a reduced risk of kidney stones in green tea consumers, but not in other types of tea consumers [18]. As for alcohol consumption, some studies have found that drinking alcohol could reduce the risk of kidney stones [14,15,17]. However, when considering different alcoholic beverage types, previous studies showed benefits for beer consumption, inconsistent results for wine consumption, and no association for spirits consumption [16,17]. Due to the limited sample size and heterogeneity in the popular drinks across populations, uncertainty remains for the associations of tea and alcohol consumption with the risk of kidney stones.

In the current study, we aimed to investigate the individual and combined associations of tea, alcohol, and fruit consumption with the risk of kidney stones in 0.5 million Chinese adults from the China Kadoorie Biobank (CKB). We especially distinguished green tea from other teas and strong spirits from other alcoholic beverages.

## 2. Materials and Methods

### 2.1. Study Population

The CKB aims to evaluate the effects of genetic and environmental factors on common chronic diseases in the Chinese population. Detailed introductions of the CKB study design and characteristics of participants have been previously described [19,20]. Briefly, from 2004 to 2008, 512,725 participants aged 30–79 years were recruited from five urban and five rural regions of China. All participants provided written informed consent and completed a laptop-based questionnaire survey, physical measurement, and blood sample collection. The study was approved by the Ethical Review Committee of the Chinese Centre for Disease Control and Prevention (Beijing, China) and the Oxford Tropical Research Ethics Committee, University of Oxford (UK).

In the present study, we excluded participants who self-reported having a history of chronic kidney disease (*n* = 7575) and cancer (*n* = 2578). We also excluded participants with missing values on body mass index (BMI, *n* = 2). The final analysis included 502,621 participants.

### 2.2. Assessment of Tea, Alcohol, and Fruit Consumption

At baseline survey, all participants were required to report their frequency of tea consumption during the past 12 months (never, only occasionally, only at certain seasons, every month but less than weekly, or at least once a week). Participants who consumed tea at least once a week were further asked to report (1) days consuming in a typical week (1 to 2 days, 3 to 5 days, or almost every day); (2) type of tea consumed most commonly (green tea, oolong tea, black tea, or others); (3) the number of cups (in 300 mL size) of tea consumed in a drinking day. A pictorial guide was provided to illuminate the standard-sized cup. According to reported tea consumption frequency, all participants were divided into three groups: never, less than daily (including only occasionally, only during certain seasons, every month but less than weekly, and weekly but less than daily), or daily. According to the cups of tea consumed per drinking day, daily consumers were further divided into four groups: 1–2, 3–4, 5–6, or ≥7 cups per day.

Similar to collecting information about tea consumption, all participants were asked to report their frequency of alcohol consumption during the past 12 months (never, only occasionally, only at certain seasons, every month but less than weekly, or at least once a week). Participants who drank alcohol once a week were further asked to report: (1) days consuming in a typical week (1 to 2 days, 3 to 5 days, or almost every day); (2) type of alcohol consumed most commonly (beer, rice wine, wine, weak spirits (<40% alcohol content), or strong spirits (≥40% alcohol content)); (3) the volume of alcohol consumed in a drinking day. The beer was calculated according to 250 mL per small bottle and 640 mL per large bottle; other types of alcohol were calculated in units of 1 Liang or 50 mL. We consulted the literature to determine the content of pure alcohol of different types of alcohol and calculate the amount of pure alcohol consumed [21]. All participants were divided into three groups based on the reported alcohol consumption frequency: never, less than daily (including only occasionally, only during certain seasons, every month but less than weekly, and weekly but less than daily), or daily. According to the amount of pure alcohol consumed in a drinking day, daily consumers were further divided into four groups: <30.0, 30.0–59.9, 60.0–89.9, or ≥90.0 g per day. According to the volume of alcohol consumed per drinking day, daily consumers were also further divided into four groups: 0–100, 101–200, 201–300, or >300 mL per day.

Habitual fresh fruit consumption was assessed by a validated qualitative food frequency questionnaire [22]. All participants were required to report how often they had consumed fresh fruits during the past 12 months (never, every month but less than weekly, 1–3 days per week, 4–6 days per week, or every day). The options “never” or “every month but less than weekly” were defined as “less than weekly”, and “1–3 days per week” or “4–6 days per week” were defined as “weekly”.

### 2.3. Assessment of Covariates

Trained investigators collected information including socio-demographic characteristics (age, sex, education, occupation, marital status, and household income), lifestyle and dietary habits (tobacco smoking, physical activity, intake of red meat, dairy products, and fresh vegetables, and dietary supplement intake of vitamins, calcium, iron, or zinc), personal medical and medication history (hypertension, diabetes, coronary heart disease, stroke, and gallstones or cholecystitis; diuretic use), and women’s menopausal status. Daily level of physical activity was calculated by multiplying the metabolic equivalent tasks (METs) value for a particular type of physical activity by hours spent on that activity per day and summing the MET hours for all activities.

Body height, weight, and waist and hip circumferences were measured with uniformly calibrated instruments. BMI was calculated by dividing weight by the square of height, and waist-to-hip ratio by the ratio of waist circumferences to hip circumferences. Prevalent hypertension at baseline was defined as measured systolic blood pressure ≥140 mmHg, and measured diastolic blood pressure ≥90 mmHg, having a self-reported diagnosis of hypertension, or using medication for hypertension. Prevalent diabetes was defined as measured fasting blood glucose ≥7.0 mmol/L, measured random blood glucose ≥11.1 mmol/L, or a self-reported diagnosis of diabetes.

### 2.4. Assessment of Outcomes

Information about morbidity and mortality of all participants during follow-up was ascertained periodically through linkage, via unique national identification number, to national health insurance (HI) claim databases, to local death and disease registries, and active follow-up. Trained staff who were blind to participants’ baseline information coded the diagnosis of disease and the cause of death with the 10th revision of the International Classification of Diseases (ICD-10). In the present study, outcome events included the first documented calculus of kidney and ureter (N20) or unspecified renal colic (N23) during the follow-up period.

### 2.5. Statistical Analysis

Baseline characteristics of participants between different consumption frequency groups of tea, alcohol, or fruits were compared by using either covariance analysis (for continuous variables) or logistic regression (for categorical variables), adjusting for age, sex, and study regions. Follow-up time (person years) was calculated from baseline survey to the date of the outcome event, death, loss to follow up, or 31 December 2017, whichever occurred first.

The Cox proportional hazard regression models, with age as the underlying time scale and stratification jointly by age (in a 5-year interval) and 10 study regions, were used to estimate hazard ratios (HRs) and 95% confidence intervals (CIs) for the associations of tea, alcohol, and fruit consumption with the risk of kidney stones. The proportional hazard assumption for the Cox regression models was tested using the Schoenfeld residual, and no violation was discovered. The multivariable models were adjusted for sex; education; occupation; household income; smoking status; physical activity; intake of red meat, dairy products, and fresh vegetables, dietary supplement intake of vitamins, calcium, iron, or zinc; menopausal status (only for women); BMI, waist-to-hip ratio; and prevalent hypertension and diabetes. The association analysis of tea, alcohol, or fruit consumption with stone risk was further mutually adjusted for the other two variables.

Tests for linear trend were conducted in daily tea or alcohol consumers by assigning the median value of tea (in cups per day) or alcohol (in grams or milliliters per day) drinking to each of the categories and including in the regression models as continuous variables. The linear trend test for fruit consumption was conducted in all participants by assigning the midpoint value of fruit consumption frequency to each of the categories.

To test the robustness of our findings, we also conducted the following sensitivity analyses: (1) excluding participants who developed kidney stones during the first two years of follow-up; (2) excluding the cases of unspecified renal colic (ICD-10: N23) during the follow-up period; (3) additionally adjusting for marital status, self-reported clinician diagnosis of coronary heart disease, stroke, and gallstones or cholecystitis, and diuretic use. The results were not substantially changed (data not shown).

We further examined the association of tea or alcohol consumption with kidney stone risk according to the types of tea or alcohol consumed most commonly. In addition, we examined the association of tea, fruits, or overall intake with kidney stone risk according to the amount of alcohol consumed. Furthermore, subgroup analyses were conducted to test whether the associations between exposure factors and risk of kidney stones were consistent in different baseline subgroups: age (<50, 50–59, or ≥60 years), gender (male or female), BMI (<24 or ≥24 kg/m^2^), tea consumption (<3 or ≥3 cups/day), alcohol consumption (<30.0 or ≥30.0 g/day), and fruit consumption (<4 or ≥4 days/week). We implemented likelihood ratio tests for the interaction, comparing models with or without interaction terms.

We used Stata, version 15.0 (StataCorp, College Station, TX, USA), for statistical analyses. All *p*-values were two-sided and statistical significance was defined as *p* < 0.05.

## 3. Results

Of all the participants included, the mean age was 52.0 ± 10.7 years, 58.9% were women, and 44.0% lived in urban areas (Table 1). The preferred types of tea and alcohol were green tea and strong spirits, accounting for 85.9% of daily tea consumers and 51.3% of daily alcohol consumers, respectively. The daily tea and alcohol consumers were more likely to be male, while the daily fruit consumers were more likely to be female. The habits of tea and alcohol drinking tended to cluster; that is, daily tea consumers were more inclined to drink alcohol every day. Daily fruits consumers were more likely to drink tea, but less likely to drink alcohol.

During a median of 11.1 years (5.4 million person years) of follow-up, we documented 12,407 cases of kidney stones. Compared with participants who never drank tea in the past year, those who drank tea ≥3 cups per day had a linearly reduced risk of kidney stones (*p* for trend < 0.001) (Table 2). The multivariable-adjusted HR (95% CI) for those who drank tea ≥7 cups per day was 0.73 (0.65–0.83). The negative association between tea consumption and risk of kidney stones was found in both green tea and other types of tea consumers (Table 3).

For alcohol consumption, the multivariable-adjusted HR (95% CI) was 0.79 (0.72–0.87) for participants who drank 30.0–59.9 g of pure alcohol per day compared to those who never drank alcohol in the last year (Table 2). However, the risk did not further decrease with the increase of alcohol consumption (*p* for trend = 0.346). The reduced risk of kidney stones was consistently observed in both weekly and daily strong spirits consumers (*p* for trend = 0.360), but not in other types of alcohol consumers (Table 4).

Fruit consumption was also associated with a lower risk of kidney stones, which further reduced as the frequency of fruit consumption increasing (*p* for trend < 0.001) (Table 2). The multivariable-adjusted HR (95% CI) for daily fruit consumers was 0.81 (0.75–0.87) compared to less-than-weekly consumers.

There was no significant difference between adequate intake of tea, fruits, or overall intake and kidney stone risk in different alcohol consumption groups (all *p*_int_ > 0.05) (Figure 1). However, the association was only statistically significant in participants who drank <30.0 g of pure alcohol per day. Compared with participants who consumed neither tea nor fruits adequately, the HRs (95% CIs) for those with intake of either and both adequately were 0.88 (0.85–0.92) and 0.73 (0.66–0.81), respectively.

The associations between exposure factors and risk of kidney stones were consistent across most of the subgroups according to age, gender, BMI, and consumption of tea, alcohol, or fruits (all *p*_int_ > 0.05) (Appendix A). The inverse association between alcohol consumption and risk of kidney stones was more pronounced in the 50–59 age group (HR = 0.73, 95% CI: 0.64–0.82) than other age groups (*p*_int_ = 0.033) (Appendix A). In addition, the inverse association between fruit consumption and stone risk was more pronounced in BMI < 24 kg/m^2^ group (HR = 0.83, 95% CI: 0.78–0.90) than in BMI ≥ 24 kg/m^2^ group (HR = 0.91, 95% CI: 0.85–0.98) (*p*_int_ = 0.001) (Appendix A).

## 4. Discussion

In this large prospective cohort study of Chinese adults, consumption of ≥3 cups of tea (about 900 mL) per day, consumption of ≥30.0 g of pure alcohol per day, and consumption of at least 4 d/w of fruit were all associated with reduced risk of kidney stones. Increasing tea and fruit consumption could independently decrease the risk of kidney stones, even for those who did not drink alcohol excessively.

The study based on three ongoing cohorts (HPFS, NHSI, and NHSII) reported a declined risk of kidney stones among participants who drank ≥1 glass (about 237 mL) of tea per day compared with those drinking <1 glass per week (HR = 0.89, 95% CI: 0.82–0.97) [16]. Results from the UK Biobank of 439,072 participants showed that only those who drank ≥5 cups (average 839 mL) per day had a statistically significantly lower risk of kidney stones (HR = 0.85, 95% CI: 0.78–0.92), compared with non-daily tea consumers [15]. Unlike the studies above, the association between tea consumption and kidney stone risk was not observed in another prospective study of 27,001 Finnish male smokers, in which the proportion of tea consumers was relatively low [17]. Our study adds to the evidence that for participants who drank ≥3 cups (about 900 mL) of tea per day, the risk of kidney stones further reduced as tea consumption increased. The reduced risk was consistently seen in people who drank green tea and other teas. The Shanghai Men’s and Women’s Health Study reported that the reduced risk of kidney stones associated with tea consumption was only seen among green tea consumers (men: HR = 0.78, 95% CI: 0.69–0.88; women: HR = 0.84, 95% CI: 0.74–0.95), possibly due to the limited number of cases among other tea consumers [18]. The caffeine in tea has a diuretic effect and could also promote sodium, chloride, calcium, phosphate, magnesium, and citrate excretion [23,24]. However, some studies have found that decaffeinated coffee might also reduce the stone risk, suggesting that the protective effect may not only come from caffeine but also from other chemicals or fluid itself [16,25]. Moreover, tea is rich in polyphenols and other various phytochemicals, which may provide protective effects against oxalate-induced toxicity [26,27,28].

A study based on Oxford data of the EPIC found that only participants who consumed ≥16 g of alcohol per day had a lower risk of kidney stones compared to those who consumed 1–7 g of alcohol per day (HR = 0.65, 95% CI: 0.47–0.91) [14]. Findings from the UK Biobank demonstrated that the risk of kidney stones decreased linearly as the volume of alcohol consumed increased (*p* for trend <0.001); the HR (95%CI) for per 200 mL/d alcohol was 0.85 (0.82–0.88) [15]. However, in the combined analysis of HPFS, NHSI, and NHSII, the reduced stone risk was only observed for consumption of >1 serving of beer and wine per day compared with drinking <1 serving per week, but not for liquor consumption [16]. Our study revealed that alcohol consumption, as high as 90.0 g/d of pure alcohol or above, could reduce the stone formation risk, but there was no linear decreasing trend. Given a large number of strong spirits consumers in our study population, we provide powerful evidence that strong spirits consumption could also reduce kidney stone risk. Alcohol is thought to have a diuretic effect by inhibiting vasopressin secretion and may further prevent the formation of stones [29]. However, there are also concerns that alcohol might increase stone risk by promoting the formation of uric acid metabolites [30,31] and lead to lithogenesis by causing oxidative stress damage to kidney tissue [32]. Nevertheless, the results of the current study did not support the latter hypothesis.

Previous studies in Western populations have shown that kidney stone risk decreased with increasing daily fruit intake, but no further risk reduction was found for much higher intake [12,14,15]. A study from the UK Biobank found that those who ate two or more servings of fruits per day had a lower risk of kidney stones, the HRs (95% CIs) for those who ate two, three, four, and five or more servings per day were 0.90 (0.83–0.98), 0.81 (0.73–0.90), 0.74 (0.63–0.86), and 0.75 (0.65–0.86), respectively, when compared with those who ate one or less serving per day [15]. In our study, the percentage of daily fruit consumers was relatively low [33]. We found that the risk of kidney stones also decreased with increased days per week of fruit consumption. Fruits are not only high in water but also rich in vitamins, inorganic salts, and fiber, which could increase urinary potassium, magnesium, citrate, and other stone inhibitors, thereby reducing the risk of kidney stones [12,34].

Given the other negative health effects of alcohol consumption, it is not advisable to reduce the risk of kidney stones by increasing the amount of alcohol consumed [35]. Our further analysis found that tea and fruit consumption could independently reduce the kidney stone risk, even for participants who did not drink alcohol excessively.

As far as we know, the present study is so far the largest study to explore the associations between tea, alcohol, and fruit consumption and the risk of kidney stones among Chinese adults. The strengths of our study included its prospective study design and the inclusion of geographically diverse participants. Due to the large sample size and long follow-up time, we accumulated enough cases to observe the trend of kidney stone risk with different levels of beverage or fruit consumption. We specifically obtained robust results for green tea and other teas and for strong spirits and other alcoholic beverages.

Inevitably, our study also has some limitations. First, we did not collect total fluid intake, types and portion size of fruit consumption, and more specific dietary factors, especially water and other beverage intakes for non-tea consumers. Even so, we still observed that an additional increase in the intake of tea, alcohol, and fruits reduced the risk of kidney stones. Second, we did not collect the history of kidney stones for participants at baseline, leading to uncertainty of whether the cases experienced the first occurrence. However, we have excluded participants with self-reported chronic kidney disease, which, to some extent, might exclude recurrent symptomatic kidney stone patients. Third, information about tea, alcohol, and fruit consumption was self-reported. Nevertheless, the recall bias might lead to non-differential misclassification and drive the associations to be null. Fourth, we only had baseline information for exposures and covariates, and the one-time estimation could not reflect the changes in lifestyle characteristics over time.

## 5. Conclusions

In a large cohort of Chinese adults, increasing consumption of tea, alcohol, or fruits was associated with a lower risk of kidney stones. Further studies are needed to clarify whether tea, alcohol, and fruit consumption reduces the stone risk through certain chemical substances, or simply by increasing the amount of fluid intake. Given the other adverse health effects of alcohol consumption, encouraging tea and fruit consumption to increase the amount of fluid intake, as well as some potentially beneficial ingredients, is a priority for simple and effective ways to prevent kidney stones.

## Figures and Tables

**Figure 1 nutrients-13-01119-f001:**
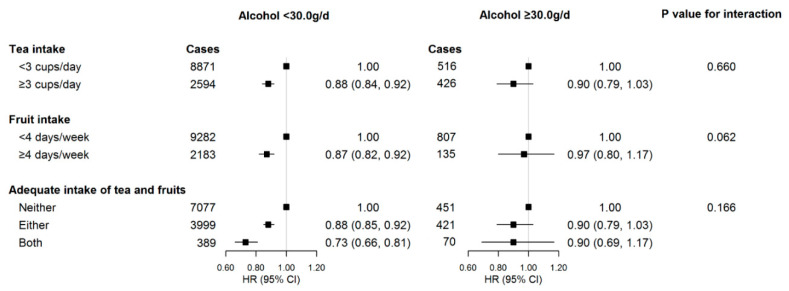
Subgroup analyses of the associations between adequate intake of tea and fruits and risk of kidney stones according to the amount of alcohol consumed (*n* = 502,621). Abbreviations: HR, hazard ratio; CI, confidence interval. Adjusted covariates were the same as those in model 3 of Table 2, as appropriate. Adequate intake was defined as tea consumption ≥3 cups/day or fruit consumption ≥4 days/week. Solid squares represent point estimates, and horizontal lines represent 95% confidence intervals.

**Table 1 nutrients-13-01119-t001:** Baseline characteristics of study participants according to the frequencies of tea, alcohol, or fruit consumption.

Variables	Tea Consumption				Alcohol Consumption				Fruit Consumption		
	Never	Less than Daily	Daily		Never	Less than Daily	Daily		Less than Weekly	Weekly	Daily
Participants, *n* (%)	178,015 (35.4)	192,565 (38.3)	132,041 (26.3)		239,402 (47.6)	217,769 (43.3)	45,450 (9.0)		203,728 (40.5)	205,162 (40.8)	93,731 (18.7)
Age, year	53.2	50.3	52.8		53.1	50.4	53.7		52.8	51.3	51.6
Women, %	76.8	58.1	35.9		79.3	47.5	6.0		55.2	58.2	68.3
Urban area, %	43.9	46.2	40.9		40.9	46.8	47.1		24.1	45.5	83.9
Middle school and higher, %	43.7	51.3	52.9		46.7	52.7	45.0		42.5	50.6	61.3
Agricultural and industrial workers, %	57.5	56.0	54.4		55.2	56.2	60.3		61.7	54.4	47.2
Household income ≥10,000 CNY/year, %	67.7	73.7	75.3		69.9	73.8	71.9		65.9	74.3	81.9
Current smoking ^a^, %											
Men	56.8	63.8	76.8		61.9	66.0	78.7		73.2	65.1	58.6
Women	2.4	2.7	4.7		2.2	3.2	8.7		3.6	2.7	1.9
Physical activity, MET-h/d	21.4	21.6	20.3		20.7	21.5	22.1		21.6	21.1	20.3
Average weekly consumption ≥4 days, %											
Red meat	44.2	47.6	49.7		45.1	48.2	53.5		40.9	48.7	55.4
Dairy products	9.9	12.1	14.3		10.4	13.2	10.4		5.8	9.6	21.0
Fresh vegetables	98.1	98.2	98.8		98.2	98.6	97.9		97.0	99.1	99.4
Fresh fruits	24.1	29.0	32.4		26.5	30.9	22.9		-	-	-
Daily tea consumption, %											
Men	-	-	-		36.2	39.3	50.6		39.9	40.7	44.7
Women	-	-	-		14.8	18.8	30.5		13.7	15.6	20.7
Daily alcohol consumption, %											
Men	14.6	17.5	26.4		-	-	-		25.2	18.6	15.7
Women	0.6	0.9	2.6		-	-	-		0.9	0.8	1.1
Use of dietary supplement ≥1 month, %											
Vitamins	3.3	4.4	4.5		3.5	4.8	3.0		2.1	3.9	7.6
Calcium, iron, or zinc	6.0	7.8	7.5		6.3	8.1	6.7		5.0	6.8	11.4
Body mass index, kg/m^2^	23.4	23.7	23.9		23.7	23.7	23.5		23.4	23.8	23.8
Waist-to-hip ratio	0.879	0.882	0.885		0.882	0.880	0.887		0.883	0.883	0.878
Hypertension, %	34.6	34.6	36.8		36.8	32.4	39.5		36.0	35.2	33.3
Diabetes, %	5.7	5.8	6.2		6.9	5.0	4.5		7.4	5.5	4.3

Abbreviations: MET, metabolic equivalent of task. All variables were adjusted for age, sex, and study regions, as appropriate. ^a^ Current smokers included former smokers who had stopped smoking because of illness.

**Table 2 nutrients-13-01119-t002:** HRs (95% CIs) for associations between tea, alcohol, and fruit consumption and risk of kidney stones (*n* = 502,621).

	Cases	Cases/PYs (/10,000)	Model 1	Model 2	Model 3
Tea consumption					
Never	2676	13.99	1.00	1.00	1.00
Less than daily	5240	25.29	0.99 (0.94, 1.04)	0.98 (0.93, 1.03)	0.99 (0.94, 1.04)
Daily (cups/day)					
1–2	1471	43.59	1.01 (0.94, 1.09)	1.01 (0.94, 1.08)	1.02 (0.95, 1.10)
3–4	2010	40.47	0.90 (0.84, 0.96)	0.89 (0.83, 0.96)	0.91 (0.85, 0.98)
5–6	660	20.66	0.88 (0.80, 0.96)	0.86 (0.79, 0.94)	0.89 (0.81, 0.98)
≥7	350	14.12	0.72 (0.64, 0.80)	0.71 (0.63, 0.79)	0.73 (0.65, 0.83)
*p* for trend ^a^			<0.001	<0.001	<0.001
Alcohol consumption					
Never	6146	24.03	1.00	1.00	1.00
Less than daily	5000	21.29	0.97 (0.93, 1.01)	0.97 (0.93, 1.02)	0.98 (0.94, 1.02)
Daily (grams/day)					
<30.0	319	21.60	0.94 (0.83, 1.05)	0.94 (0.84, 1.06)	0.95 (0.85, 1.07)
30.0–59.9	529	26.42	0.79 (0.72, 0.87)	0.78 (0.71, 0.86)	0.79 (0.72, 0.87)
60.0–89.9	177	28.80	0.78 (0.67, 0.91)	0.76 (0.65, 0.89)	0.77 (0.66, 0.90)
≥90.0	236	33.23	0.77 (0.67, 0.88)	0.75 (0.66, 0.86)	0.77 (0.67, 0.88)
*p* for trend ^a^			0.609	0.328	0.346
Fruit consumption					
Less than weekly	5533	25.65	1.00	1.00	1.00
1–3 d/w	4556	26.82	0.95 (0.92, 0.99)	0.96 (0.93, 1.01)	0.96 (0.92, 1.00)
4–6 d/w	1028	20.10	0.89 (0.83, 0.95)	0.90 (0.84, 0.97)	0.89 (0.83, 0.96)
7 d/w	1290	12.66	0.77 (0.72, 0.83)	0.81 (0.75, 0.87)	0.81 (0.75, 0.87)
*p* for trend ^b^			<0.001	<0.001	<0.001

Abbreviations: HR, hazard ratio; CI, confidence interval; PYs, person years. Model 1 was adjusted for sex (male, female); model 2: further adjusted for education (no formal school, primary school, middle school, high school, college, or university or above), occupation (agriculture, industrial, administrative or managerial, professional or technical, sales or service, retired, house wife or husband, self-employed, unemployed, or other), household income (<2500, 2500–4999, 5000–9999, 10,000–19,999, 20,000–34,999, or ≥35,000 CNY/year), smoking status (never or occasional, former, daily smoking 1–14, 15–24, or ≥ 25 cigarettes or equivalent tobacco; former smokers who had stopped smoking because of illness were included in the current daily smokers), physical activity (MET-h/day), intake of red meat, dairy products, and vegetables (variables were assigned according to the midpoint value of intake frequency: never or rarely = 0, monthly = 0.5, 1–3 days/week = 2, 4–6 days/week = 5, or daily = 7; and were adjusted as continuous variables), dietary supplement use of vitamin, calcium, iron or zinc for at least one month (yes or no), BMI (kg/m^2^), waist-to-hip ratio, prevalent hypertension (presence or absence), and prevalent diabetes (presence or absence); model 3: further mutually adjusted for tea, alcohol, and fruit consumption. ^a^ Tests for linear trend were conducted in daily tea or alcohol consumers by assigning the median value of tea (in cups per day) or alcohol (in grams per day) drinking to each of the categories. ^b^ Test for linear trend were conducted in all participants by assigning the midpoint value of fruit intake frequency to each of the categories.

**Table 3 nutrients-13-01119-t003:** Subgroup analyses of association between tea consumption and risk of kidney stones according to types of tea consumed most commonly.

	Green Tea (*n* = 321,376)			Other Types of Tea ^a^ (*n* = 202,985)		
	Cases	Cases/PYs (/10,000)	HR (95% CI)	Cases	Cases/PYs (/10,000)	HR (95% CI)
Never	2676	13.99	1.00	2676	13.99	1.00
Weekly ^b^	1047	32.60	0.96 (0.88, 1.05)	112	16.10	1.00 (0.82, 1.23)
Daily (cups/day)						
1–2	1406	46.85	1.00 (0.92, 1.09)	65	17.39	0.99 (0.77, 1.29)
3–4	1903	45.17	0.89 (0.82, 0.96)	107	14.18	0.79 (0.64, 0.98)
5–6	591	21.22	0.87 (0.78, 0.96)	69	16.88	0.84 (0.65, 1.08)
≥7	280	13.99	0.73 (0.63, 0.83)	70	14.66	0.62 (0.48, 0.81)
*p* for trend ^c^			<0.001			0.047

Abbreviations: HR, hazard ratio; CI, confidence interval; PYs, person years. Adjusted covariates in the models were consistent with the model 3 in Table 2. ^a^ Other types of tea included oolong tea, black tea, white tea, and so on. ^b^ Participants who consumed tea only occasionally, only at certain seasons, or monthly but less than weekly were not included in the analysis because they were not asked to report the commonly consumed tea type. ^c^ Tests for linear trend were conducted in daily drinkers by assigning the median value of tea drinking (in cups per day) to each of the categories as continuous variables in regression models.

**Table 4 nutrients-13-01119-t004:** Subgroup analyses of the association between alcohol consumption and risk of kidney stones according to types of alcohol consumed most commonly.

	Strong Spirits (*n* = 274,421)			Other Types of Alcohol ^a^ (*n* = 279,198)		
	Cases	Cases/PYs (/10,000)	HR (95% CI)	Cases	Cases/PYs (/10,000)	HR (95% CI)
Never	6146	24.03	1.00	6146	24.03	1.00
Weekly ^b^	487	39.04	0.89 (0.81, 0.99)	258	13.31	0.97 (0.84, 1.12)
Daily (mL/day)						
0–100	406	38.86	0.84 (0.76, 0.94)	65	13.70	0.96 (0.74, 1.23)
101–200	310	36.16	0.71 (0.63, 0.81)	79	14.30	0.90 (0.71, 1.14)
201–300	167	40.91	0.77 (0.65, 0.91)	74	15.69	0.95 (0.75, 1.21)
>300	43	35.50	0.68 (0.50, 0.92)	117	13.39	0.92 (0.76, 1.13)
*p* for trend ^c^			0.360			0.580

Abbreviations: HR, hazard ratio; CI, confidence interval; PYs, person years. Adjusted covariates in the models were consistent with model 3 in Table 2. ^a^ Other types of alcohol included beer, rice wine, wine, and weak spirits. ^b^ Participants who consumed alcohol only occasionally, only during certain seasons, or monthly but less than weekly were not included in the analysis because they were not asked to report the commonly consumed alcohol type. ^c^ Tests for linear trend were conducted in daily drinkers by assigning the median value of alcohol drinking (in milliliters per day) to each of the categories as continuous variables in regression models.

## Data Availability

The access policy and procedures are available at www.ckbiobank.org (accessed on 12 March 2021).

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
