# Peer review of "Consumption of Tea, Alcohol, and Fruits and Risk of Kidney Stones: A Prospective Cohort Study in 0.5 Million Chinese Adults"

_nutrients, 2021, doi:10.3390/nu13041119_

Round 1
Reviewer 1 Report
In this paper the authors prospectively look at the consumption of tea, alcohol and fruits with the risk of kidney stone disease for a cohort of 0.5 million Chinese adults.
This is an excellent study which has looked at a large number of patients in a prospective manner and within the limitations that data on fluid intake is self-reported and previous history of stones is not taken account of, it is still able to show that increasing consumption of tea, alcohol, or fruits was associated with lower risk of kidney stones.
Under the limitations, they must also add that the type of fruit taken and the portion size was not known. They should also add the association of kidney stones with metabolic syndrome.
Also, they are missing certain important references which needs to be added –
- Tea and coffee consumption and pathophysiology related to kidney stone formation: a systematic review. Barghouthy World J Urol 2020 Oct 14. doi: 10.1007/s00345-020-03466-8.
- The role of fluid intake in the prevention of kidney stone disease: A systematic review over the last two decades. Gamage KN. Turk J Urol 2020 Nov;46(Supp. 1):S92-S103.doi: 10.5152/tud.2020.20155.
- The association of metabolic syndrome and urolithiasis. Wong YV. Int J Endocrinol 2015;2015:570674. doi: 10.1155/2015/570674.Epub 2015 Mar 22.
Reviewer 2 Report
Thank you for the opportunity to review this manuscript. The authors' described their analysis of diet and beverage consumption with risk of kidney stone incidence in the CKB. They have clearly described their rationale, methodologies, and statistical approach, and interpreted the results with appropriate detail and consideration of related literature. I believe the paper can be accepted in its current form.
